# Circa-SCOPE: high-throughput live single-cell imaging method for analysis of circadian clock resetting

Gal Manella [1], Dan Aizik[1], Rona Aviram[1], Marina Golik[1] & Gad Asher [1 ✉]

Circadian clocks are self-sustained and cell-autonomous oscillators. They respond to various extracellular cues depending on the time-of-day and the signal intensity. Phase Transition Curves (PTCs) are instrumental in uncovering the full repertoire of responses to a given signal. However, the current methodologies for reconstructing PTCs are low-throughput, laborious, and resource- and time-consuming. We report here the development of an efficient and high throughput assay, dubbed Circadian Single-Cell Oscillators PTC Extraction (Circa-SCOPE) for generating high-resolution PTCs. This methodology relies on continuous monitoring of single-cell oscillations to reconstruct a full PTC from a single culture, upon a one-time intervention. Using Circa-SCOPE, we characterize the effects of various pharmacological and blood-borne resetting cues, at high temporal resolution and a wide concentration range. Thus, Circa-SCOPE is a powerful tool for comprehensive analysis and screening for circadian clocks' resetting cues, and can be valuable for basic as well as translational research.

---

[1] Department of Biomolecular Sciences, Weizmann Institute of Science, 7610001 Rehovot, Israel. ✉email: gad.asher@weizmann.ac.il

In mammals, circadian clocks are present in most cells of the body and function in a self-sustained and cell-autonomous manner[1,2]. Concomitantly, these clocks are constantly aligned with the environment through external timing cues (a.k.a. zeitgebers)[3]. Misalignment between circadian clocks and the environment is associated with various pathologies[4,5], thus highlighting the important role of these timing cues. Circadian clocks respond to a wide variety of signals in a dose- and time-dependent manner. Studies show that both in vivo (e.g., on animals' behavioral rhythms) and in vitro (e.g., using cell culture reporter assays), various factors can cause phase-advance, phase-delay, or have no effect, depending on their intensity and time-of-exposure[6–12]. Hence, considering the interaction between these variables is critical when examining the effect of a given signal on the clock.

Phase transition curves (PTCs) (or phase response curves (PRCs)) are means of analysis and visualization of time-dependent resetting and have been highly informative in this conjunction[13–15]. PTCs represent the effect of a stimulus on the phase of the clock as a function of the time it was applied, and are essential for determining the full repertoire of responses to a given signal. PTCs have been generated for a wide variety of organisms (e.g., flies[14] and mice[16]) as well as cells in culture[7,8,10]. However, the current methodologies for reconstructing PTCs are rather low throughput, laborious and time-consuming. Typically, rhythmicity is monitored continuously to determine the phase before and after an intervention, which is given at different times throughout the circadian cycle, each time on a different set of individuals (e.g., organisms, cell culture plates), and thus require large cohorts and frequent interventions. Consequently, due to practical reasons, experimentalists often settle on low resolution, single-dose PTCs, or even a single-time exposure, which are much less informative.

In view of the above, there is growing interest in high-throughput methods that comprehensively analyze the effect of different physiological and pharmacological factors on the clock. We report here the development of an assay dubbed **Circa**dian **S**ingle-**C**ell **O**scillators **P**TC **E**xtraction (**Circa-SCOPE**), for the reconstruction of high-resolution PTCs. In this method we continuously monitor oscillations at the single-cell level and concomitantly assigning their phases before and after a one-time intervention, reconstructing complete high-resolution PTC from a single culture well. Using Circa-SCOPE, we analyzed the effect of known as well as uncharacterized pharmacological and blood-borne signals on circadian clocks at high temporal resolution and a wide range of concentrations. Specifically, we tested the resetting capacity of various steroids and estimated their relative contribution in the context of serum-induced resetting. Circa-SCOPE is, thus, a valuable and powerful tool for studying the interaction between circadian clocks and resetting cues.

## Results

**Automated long-term single-cell tracking and rhythmicity analysis**. Conventionally, to construct a PTC in cultured cells, cells are synchronized to a distinctive initial phase and hence regarded uniformly. An intervention (e.g., light exposure or serum supplementation) is applied at different times throughout the circadian cycle, each time on a different culture, and the effect on the phase of the clock is determined.

While designing Circa-SCOPE we took advantage of the finding that clocks in cultured cells are cell-autonomous and tick independently of each other[7] and therefore can be considered individually. Assuming that a non-synchronized population of cultured cells covers all initial phases, we can apply a resetting signal on this mixed-phase population, and reconstruct the entire PTC from a single culture by monitoring individual cells (Fig. 1a). Biologically, a prerequisite for such an approach to work is that the cell population contains a full representation of all phases. Technically, it requires the capability to continuously track a sufficient number of individual cells for several days, before and after the intervention, and to monitor their circadian rhythmicity throughout this time frame.

To this aim, we employed an NIH-3T3 cell line that stably expresses a nuclear fluorescent clock-reporter (NIH-3T3 Rev-VNP-1; *Reverbα-Venus-NLS-PEST*)[7] and introduced a second fluorescent reporter that is constitutively-expressed in nuclei (*CMV-H2B-mCherry*[17]) to generate NIH-3T3 NR-RVNP cells (*Nuclear Red, Rev-VNP*) (Fig. 1b). The *Rev-VNP* is used for monitoring circadian rhythmicity[7,18–20], while the *H2B-mCherry* serves for continuous tracking of single-cell nuclei. To test whether this dual reporter cell line meets the aforementioned biological and technical requirements, we employed the following experimental protocol: NIH-3T3 NR-RVNP cells were plated, their culture media was replaced on the following day, and after three additional days the cells were set in the microscope for 9 days of recording with image acquisition at 1 h intervals. The experimental procedure was followed by a customized image analysis pipeline, which segments the nuclei, tracks them, and measures their fluorescence intensity (as detailed in the Methods section). To minimize cell proliferation, which might disrupt cell tracking, cells were grown in 1% fetal bovine serum (FBS). Under these conditions, the population size is reasonably constant, and the population density does not compromise the nuclei segmentation (Fig. 1c). This stability is critical for optimizing tracking efficiency. We observed a reduction in the number of trackable cells throughout the experiment (Fig. 1d), due to biological (e.g., cell division and death) and technical reasons (e.g., cells moving out-of-frame). Overall, ~25% of the cells were fully trackable from the beginning till the end of the experiment, namely for 9 consecutive days.

Next, we examined the rhythmicity of the tracked cells. The Rev-VNP fluorescence profile of each cell was detrended and fitted with a cosine function in a 3-day window (between time-points 24–96 h). Fits with $R^2 > 0.5$ were considered rhythmic (Fig. 1e, f, see representative traces in Supplementary Fig. 1a), and they comprised ~50% out of the trackable cells throughout the experiment (Fig. 1g). To validate our fit-based rhythmicity criterion, we compared it with three other independent rhythmicity tests and found a very good agreement between them all, in terms of rhythmicity significance, and period and phase estimation (Supplementary Fig. 1b–d). Importantly, phase analysis showed a wide distribution, with a sufficient representation of all initial phases (Fig. 1h). The circadian profile of single-cells is often noisy from cycle to cycle[18,21,22], which may affect our phase estimate and consequently the PTC accuracy. To validate our phase estimate, we compared the peak time of the cosine fit in each cycle to its corresponding local maximum in the raw data, a commonly used measure in assessing the phase dynamics in single-cells[18,22]. As depicted in Supplementary Fig. 1e, f, there is no systematic bias in phase estimate in either of the cycles, and the standard deviation between the two estimates is less than 1 h in each direction. It is noteworthy that the observed deviation stems from inaccuracies in both methods (e.g., maxima-based estimations are prone to false detection due to outlier measurements).

In these conditions, we observed that the circadian period length varied between individual cells, in line with the previous reports[7,22]. It also changed throughout the experiment, gradually drifting from $26 \pm 1$ h at the beginning of the recording to $25 \pm 1$ h at the end of the experiment (Fig. 1i). The variability in the period between consecutive days might

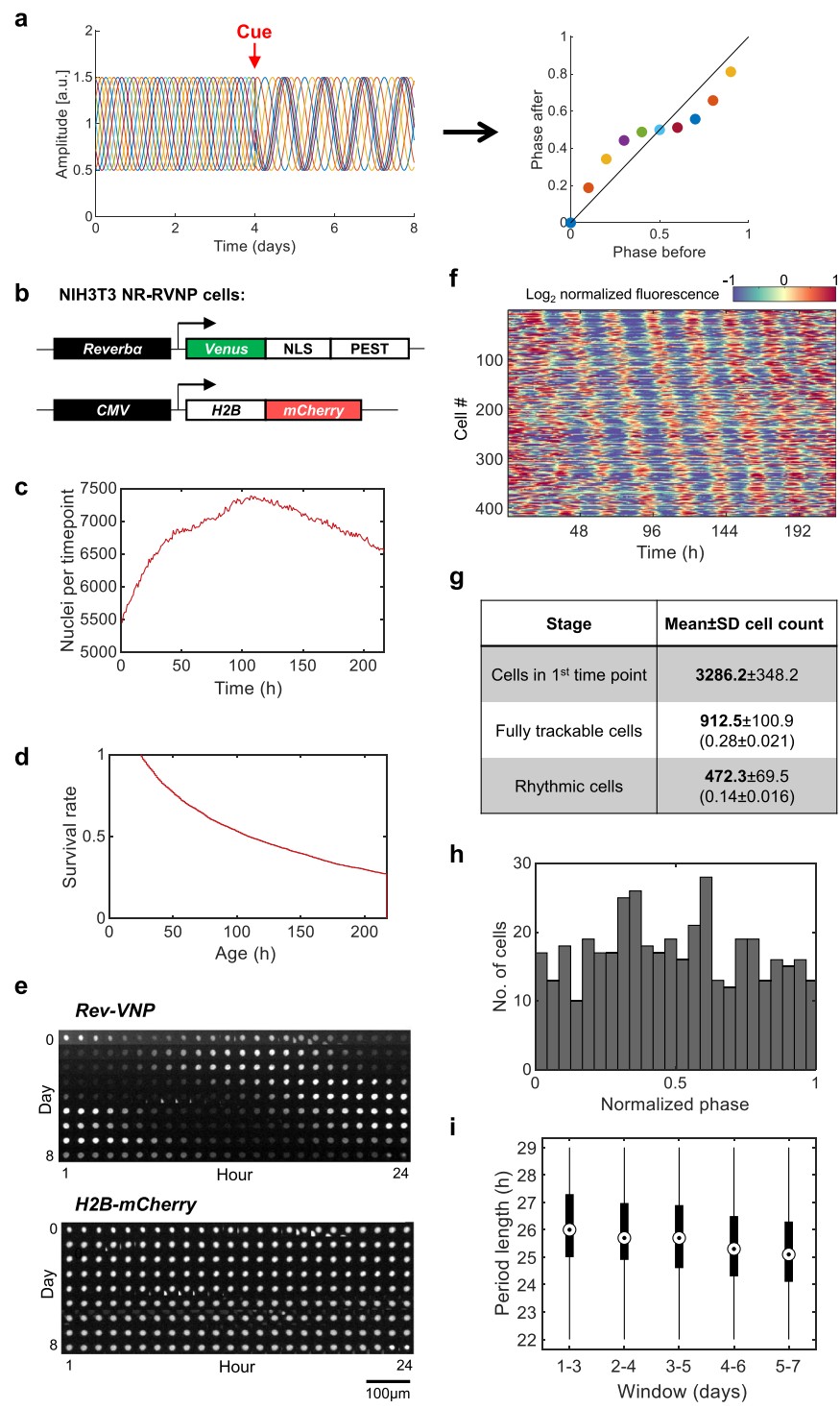

stem from a gradual change in growth conditions (e.g., nutrient content and media pH). These inherent period changes were taken into consideration in the computation of the PTC, as detailed below. Overall, our results show that the experimental and computational pipeline fulfills the necessary pre-requirements, namely traceability, rhythmicity, and comprehensive phase representation.

**High-resolution PTC reconstruction using Circa-SCOPE.** Having established a trackable cell culture setting for monitoring the circadian rhythmicity of individual cells, we set out to test the

compatibility of our system with phase resetting experiments. We employed the following experimental design (Fig. 2a): Growth media was replaced one day following cell plating. After 3 additional days, the plate was set in the IncuCyte microscope for live imaging. Images were taken continuously at 1 h intervals for 9 subsequent days. After the first 4.5 days of recording, a treatment of choice was applied. Once the experiment was completed, the cell-imaging data was retrieved and analyzed as described above to establish single-cell circadian profiles. To construct a PTC, the profiles were fitted with a cosine model in two-time windows: before (24–96 h) and after (120–192 h) the intervention ("cue"), (Fig. 2b). The fitting curves were used to determine the

**Fig. 1 Automated long-term single-cell tracking and rhythmicity analysis. a** Schematic depiction of the principle underlying Circa-SCOPE: a non-synchronized cell population is exposed to a single time intervention ("cue"). The clock in each cell is expected to respond according to its initial phase. By continuously monitoring single-cell rhythmicity, a full high-resolution PTC can be reconstructed based on data retrieved from a single population. **b** NIH-3T3 NR-RVNP cells (*Nuclear Red, Rev-VNP*) stably express the following constructs: a *Reverbα-Venus-NLS-PEST* (Rev-VNP) and *CMV-H2B-mCherry*. NLS: Nuclear Localization Signal; PEST: a proteolysis signal peptide from the mouse *Odc1* gene; H2B: Histone 2B. **c–i** Recordings of NIH-3T3 NR-RVNP cells for nine consecutive days using the IncuCyte microscope. **c** The total number of nuclei detected per time point throughout the experiment in one representative well (the data in panels **d–f** and **h–i** are retrieved from the same population). **d** The survival rate throughout the experiment of the cells detected at the first time point. **e** Example of time-lapse images of a single nucleus (out of 417 rhythmic traces) as tracked throughout the entire experiment of *Rev-VNP* (green fluorescence) and *H2B-mCherry* (red fluorescence) reporters. **f** Heatmap representation of the $\log_2$ normalized fluorescence of Rev-VNP, in all fully trackable rhythmic cells. Each row corresponds to a single cell ($n = 417$ cells in total). **g** Summary table of the cell counts after each step of the analysis. (Mean ± SD from six different wells; In brackets: the fraction from cells in first-time point). **h** Phase distribution of the rhythmic cells, normalized to the period length of each cell ($n = 417$ cells in total). **i** Period length distribution of the rhythmic cells in 3-day windows across the experiment ($n = 417$ cells. Central circle: median; box: interquartile range (IQR); whiskers: extend to the maximal/minimal values within the range of ±1.5 IQR; circles: outliers). Source data are provided as a Source Data file.

"Old phase" and the "New phase" following the intervention, which was defined by the first peak of the cosines in the second window. Since in most cases the period lengths between the two windows differed (see Fig. 1i), we implemented a simple correction named tau-independent phase analysis (TIPA)[23], which stretches or squeezes the period of the "before" fit to match the period of the "after" fit. In addition, all phases were normalized to their period and are given in relative units (i.e., between 0 and 1). Profiles with poor fits (according to $R^2$) or with extreme period length (>29 or <22) were filtered out.

As a proof of principle, we selected dexamethasone, a well-established and widely used pharmacological intervention for resetting clocks in cell culture[24]. We compared untreated with 100 nM dexamethasone-treated cell cultures (Fig. 2c, d). The filtered phases were plotted as a standard PTC, with Old Phase on the $x$-axis and New Phase on the $y$-axis, or as a PRC, where the Old Phase is on the $x$-axis and the relative phase shift is on the $y$-axis. As conventionally done - due to the circular nature of the phase coordinates - the data was double-plotted. While no phase shift was apparent in the untreated culture, the culture treated with dexamethasone showed a clear response, which was consistent with previous reports[10,25].

Broadly, phase resetting can fall within two distinct categories, as reflected in the slope of the PTC[13,15,26]. A PTC with a mean slope of one is defined as *type-1 resetting* and considered "weak" since it elicits only modest phase shifts. On the other hand, a PTC with a mean slope of zero, termed *type-0 resetting*, is regarded as "strong" as the new phase will be similar irrespective of the original phase. Other topological characteristics are informative, for example, the presence of "dead-zones", namely time windows where there is no phase shift. To objectively assign PTCs to different types and compare between their characteristics, we developed and employed the following mathematical analysis: we fitted the PTC with two Fourier models, one under the assumption of type-1 resetting and a second assuming a type-0 resetting (Fig. 2e), and applied bootstrapping-based statistical tests on them. The first test (Fig. 2f) is against the null hypothesis of no response, with a null distribution produced by resampling of the maximal phase-shifts from an untreated control PTC. The second test is employed for selection between type-1 (the null hypothesis) and type-0 resetting (Fig. 2g). The tested parameter is the root mean squared error (RMSE), an established measure for models' goodness-of-fit, whereby the lower the RMSE — the better the fit. Therefore, by resampling and fitting both type-0 and type-1 models, we constructed a distribution of the difference in RMSE between the alternative models. A negative value in more than 95% of the cases deemed the type-0 model significant, as indeed occurred in the dexamethasone example. This result is consistent with a low-resolution PTC obtained with the

conventional approach, namely treatment of synchronized cell population with dexamethasone throughout the circadian cycle (Supplementary Fig. 2)[7]. A lower concentration of dexamethasone (1 nM) elicited a type-1 response (Supplementary Fig. 3a–d). To gain more insight into the dynamics of the phase response, we compared the PTCs reconstructed using different "after" windows (Supplementary Fig. 3e–g). We observed that the overall topology of the PTC is preserved even when the selected window is further away from the cue, whereas the phase-dispersion increases gradually. This is to be expected considering the relative stochasticity of periods/phases in single cells. We also did not observe any correlation between the instantaneous amplitude just before the cue and the resultant phase shift (Supplementary Fig. 3h).

As above noted, the period length varies throughout the experiment (Fig. 1i) and hence differs between the before and after windows in untreated control culture (Fig. 2h). Dexamethasone not only affected the phase but also shortened the period length compared to the control cells (Fig. 2h). It is noteworthy that we did not observe marked differences when the PTCs were reconstructed separately for different bins of period length, indicating that the method is robust to period length changes (Supplementary Fig. 4). Thus, Circa-SCOPE is compatible with interventions that carry a combined effect on both phase and frequency.

**Circa-SCOPE uncovers diverse phase-resetting profiles.** We set out to examine the effect and characteristics of different known, alongside less established, resetting agents. As a showcase, we selected sodium bicarbonate ($Na_2CO_3$), phorbol-12-myristate-13-acetate (PMA), LiCl, and $CoCl_2$ (Fig. 3a, b). Although some evidence suggests these may interact with the clock[9,27–30], to date a full PTC was never generated for these compounds. All four compounds elicited a significant phase response, with a type-1 topology for the given concentrations (Fig. 3c). Yet, the PTCs clearly differed in the time and extent of their maximal shifts (Fig. 3d, e). Interestingly, we observed asymmetries between the maximal advance and maximal delay: bicarbonate, mostly elicited phase delay, $CoCl_2$ and LiCl also favored phase delays albeit to lesser extent, while in the case of PMA the response was fairly symmetric (Fig. 3d). The phase of maximal shift also differed between the compounds. Notably, $CoCl_2$ was almost phase-inverted compared to the other compounds, as it elicited phase advance at times where the rest caused phase delay and vice versa (Fig. 3e). Lastly, the different treatments also elicited distinct effects on the period length (Fig. 3f). Experiments performed with additional concentrations of these compounds further support their distinct behaviors (Supplementary Fig. 5). We conclude that Circa-SCOPE enables the discovery and comprehensive characterization of new resetting

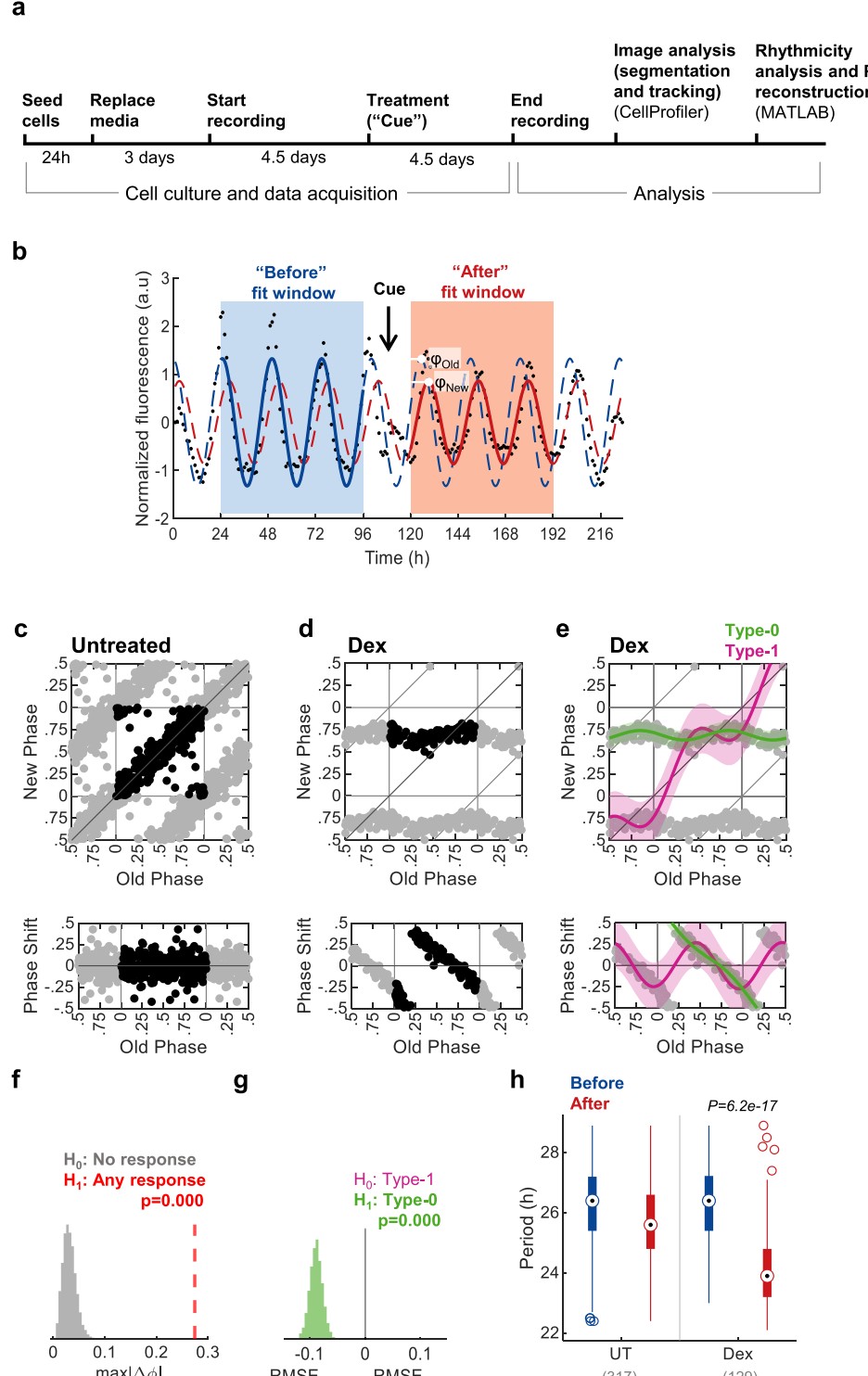

agents with versatile and intricate responses. It is conceivable that these differences stem from molecular differences in their mode of action on the clock.

**Reconstruction of dose-dependent PTCs using Circa-SCOPE.** Vast theoretical and experimental work clearly shows that the PTC topological characteristics depend on the strength of the resetting signal[10,15,31,32]. More specifically, the response to some signals can turn abruptly from type-1 to type-0 resetting when their strength is

elevated enough. In a unique point of time- and dose-of-administration lies a *phase singularity point*, a condition where the phase is mathematically indefinable. A three-dimensional PTC, which includes different doses as the third dimension, is necessary for determining the singularity point. However, using traditional methods, producing such a three-dimensional PTC is practically impossible. Circa-SCOPE appears to be ideal for such cases as it can easily and rapidly reconstruct multiple high-resolution PTCs simultaneously. As a proof-of-principle, we exposed cells to increasing doses of forskolin[10,25], from 0 to 10 μM. As predicted, we

**Fig. 2 PTC construction using Circa-SCOPE. a** Schematic depiction of Circa-SCOPE experimental protocol. **b** Exemplary rhythmic profile of a single cell throughout a resetting experiment. Dots represent raw data (in arbitrary units, a.u); the "Before" and "After" fit windows are used to produce the before (blue curve) and after (red curve) cosine fits, respectively. "Cue" marks the time of intervention; The "Old Phase" ($\varphi_{Old}$) and the "New Phase" ($\varphi_{New}$) are retrieved from the fits as denoted in the plot. **c** phase transition curve (PTC, upper panel) and phase response curve (PRC, lower panel) representations of untreated culture. Each dot corresponds to a single cell. The data is double plotted (shaded dots) for clarity. In all PTCs and PRCs, phase is given in relative units, where 1 is the cell's period length, ($n = 317$ cells). **d** PTC (upper panel) and PRC (lower panel) representations of culture treated with 100 nM dexamethasone (Dex), ($n = 129$ cells). **e** Same data as in (**d**), fitted with two Fourier-based models, one for type-1 resetting (pink) and another for type-0 resetting (green). Fits are presented ±95% confidence intervals. **f** Bootstrapping test for the response significance (one-sided). The maximal absolute phase shift of the type-1 fit of untreated control was resampled 10,000 times and compared to the measured value of dexamethasone-treated fit.
**g** Bootstrapping test for the selection between type-1 and type-0 models (one-sided). The dexamethasone-treated data was resampled 10,000 times, each time fitted with both models, and RMSE (root mean square error) was calculated for both. The model with the lower RMSE is selected, hence the p-value for type-0 is the probability of having ($RMSE_{type-0} - RMSE_{type-1}) > 0$. **h** Comparison of period length between untreated (UT) and dexamethasone-treated cultures. Two-sample two-sided Student's t-test between the period-change (After–Before) of the UT control and the period-change in the treated group; ($n = 317$ and 129 cells, respectively, as also indicated in brackets and in the legends of panels (**c**) and (**d**). Central circle: median; box: interquartile range (IQR); whiskers: extend to the maximal/minimal values within the range of ±1.5 IQR; circles: outliers). Source data are provided in Supplementary Dataset 1.

---

found a gradual increase in the magnitude of the phase-response with increasing dosage (Fig. 4a). We observed only minor effects on the period length (Supplementary Fig. 6). Remarkably, between 2 and 4 μM, the response was shifted from type-1 to type-0 (Fig. 4b). Plotting the data with a third-dimension representation of forskolin concentration, highlighted the singularity-point at a dose between 2 and 4 μM and initial phases around (-0.1)–0.1 (Fig. 4c).

Our analyses uncover the dynamics of dose-dependency resetting, namely which phase-shift parameters are dose-dependent and which are not. In the type-1 range, the maximal phase shift (both phase advance and delay) was increasing gradually in a dose-dependent manner (Fig. 4d), while in the type-0 range, as expected, the maximal phase-shift was saturated, by definition (±0.5, equivalent to ±12 h when the period length is 24 h). By contrast, the phase of maximal phase-shift remained constant throughout the entire type-1 range (Fig. 4e). Hence, it appears that the PRC amplitude is dose-dependent, while the PRC phase is dose-resilient. In the type-0 range, the mean final phase was also quite stable throughout different concentrations (Fig. 4f), whereas the range of final phases (indicated by the error bars in Fig. 4f) gradually decreased with dose. In summary, Circa-SCOPE is highly effective for the reconstruction of 3D PTCs and analysis of dose-dependent effects on clock resetting.

**Steroids PTC screen using Circa-SCOPE.** A potentially valuable application of a high-throughput PTC reconstruction methodology is drug screening. This can be done either for the discovery and characterization of drugs that can be used for clock resetting, or the identification of potential side-effects of known drugs on the clock. To demonstrate the potential of Circa-SCOPE in this regard, we performed a small screen of steroid hormones and drugs. Steroids, and especially glucocorticoids, are well-known resetting agents of peripheral clocks[24]. We tested seven different steroids: dexamethasone, prednisone, hydrocortisone, corticosterone, progesterone, β-estradiol, and testosterone with concentrations spanning 4 orders-of-magnitude (0.002–2 μM), (Supplementary Fig. 7). Within the tested dosage, β-estradiol and testosterone did not elicit a significant phase response (Fig. 5a, b), yet they affected the period length (Fig. 5c). The other steroids differed in their potency, namely the concentration of reaching type-0 resetting (Fig. 5a), and maximal phase shift per each concentration (Fig. 5b). In this regard, the most potent steroid tested was dexamethasone, with type-0 PTC observed already with concentrations as low as 0.002 μM. The different steroids did not differ much in the phase in which they induce their maximal shift (Fig. 5d), or in the mean final phase in type-0 (Fig. 5e), suggesting a common mode of action for all of them.

**Dissection of steroids role in serum-induced clock resetting.** Hitherto, we employed Circa-SCOPE to study the effect of specific purified compounds on the clock. Blood serums have been extensively used to reset clocks in cultured cells[33]. They contain a myriad of potential resetting agents; however, these factors' identities and relative contributions are not well characterized. Initially, we set out to examine the phase response characteristic of serums from different species. We tested the effect of human serum, rat serum, and FBS, as well as an artificial serum-replacement (i.e., B-27 supplement), (Supplementary Fig. 8a–d). They all induced a significant phase response, yet differed in their potency, with the rat serum and B-27 eliciting the strongest response (Supplementary Fig. 8e, f). The phase of a maximal shift in type-1 (Supplementary Fig. 8g), and the mean final phase in type-0 (when applicable, see Supplementary Fig. 8h), were comparable between the different serums. Notably, FBS and human serum induced a period shortening (Supplementary Fig. 8h). The observed differences in potency can stem from disparities in the composition of the serums, which potentially contain various resetting and inhibitory molecules.

While blood serum contains steroids[34], which are sufficient to reset the clock (Fig. 5), their role in serum-induced phase-resetting remains unclear. To directly test the potential effect of steroids on the clock in this conjunction we employed mifepristone, a potent glucocorticoid receptor (GR) and progesterone receptor (PR) antagonist[35]. We found that mifepristone completely abolished the phase-resetting effect of dexamethasone (Supplementary Fig. 9). This result encouraged us to use mifepristone to eliminate GR/PR-dependent responses, and thus qualify the significance of steroids in serum-induced resetting. We found that the clock resetting effect of serum in mifepristone-treated cells was altered compared to untreated control cells (Fig. 6a). Mifepristone reduced the phase shift strength for all tested serum concentrations, in particular for phase delays (Fig. 6b, c). In addition, the phase of maximal shift slightly differed in mifepristone-treated compared to control cells (Fig. 6d). It is noteworthy that the resetting parameters of serums (Supplementary Fig. 8) differed from those observed for steroids (Fig. 5), in particular for the mean final phase induced in type-0. This is consistent with our finding that GR/PR-dependent resetting does not account for the entire resetting effect of serum. Thus, we conclude that steroids play an important role in serum-induced resetting, yet other factors are probably implicated as well in the full response.

**Discussion**
Circadian reporters have been widely used in mammals for over two decades for studying circadian rhythms[36,37]. Many studies relied on these reporters to examine the effect of different signals

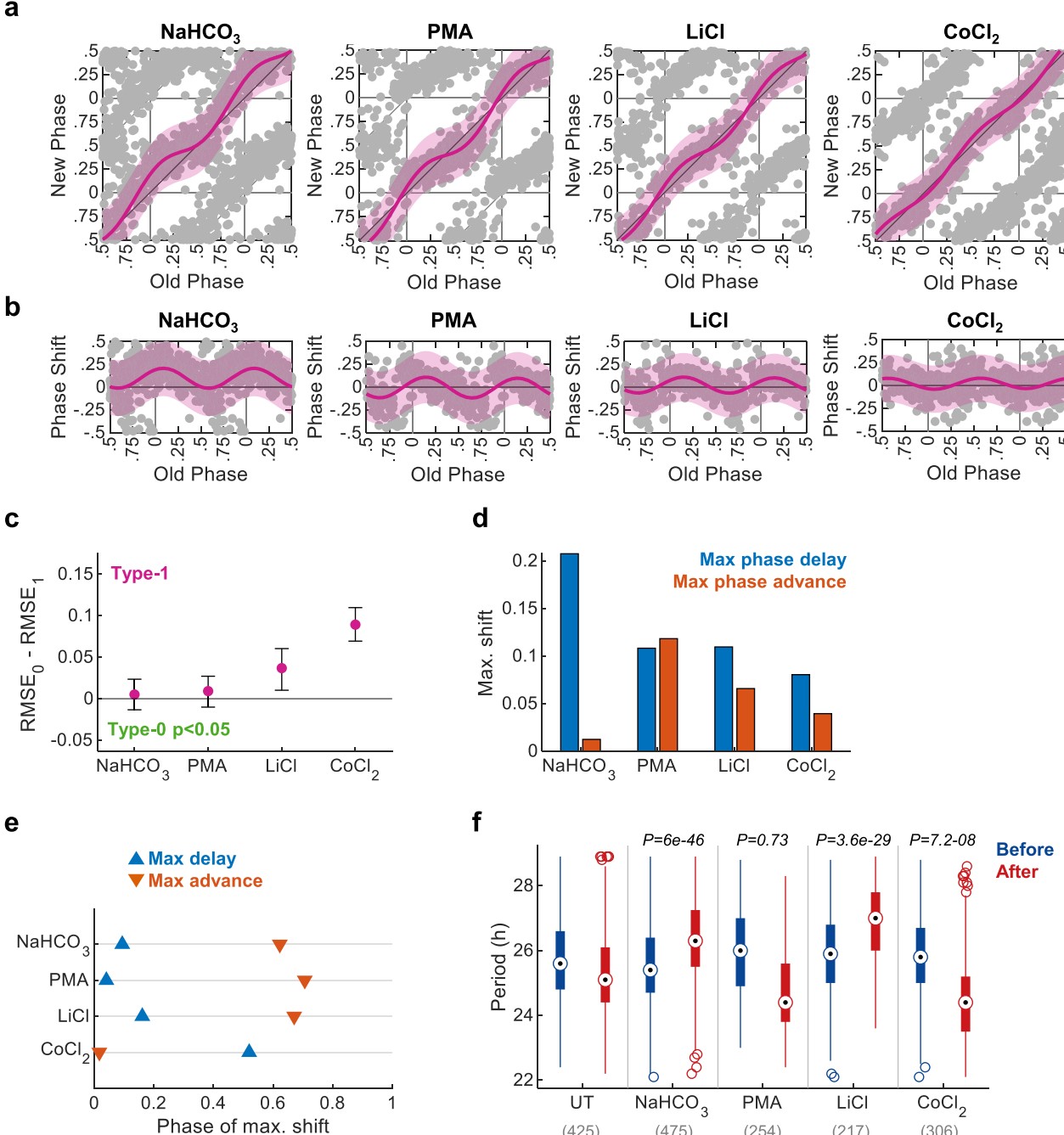

**Fig. 3 Circa-SCOPE can identify diverse PTC topologies. a** PTC and **b** PRC representation of resetting experiments conducted with 100.25 mM of bicarbonate (initial concentration in the medium was 44 mM), 4 µM PMA, 25 mM LiCl, or 80 µM of CoCl₂. Data are fitted with the type-1 model (pink) and presented ±95% confidence interval. All PTCs have significant response ($p < 0.05$ with bootstrapping test, $n = 475$, 254, 217, and 306 cells per condition respectively). **c** Model selection for each treatment. Dots represent the observed RMSE₀–RMSE₁, error bars represent the 95% confidence interval from bootstrapping test, and color indicates whether the test was significant for type-0 (one-sided). **d** Maximal phase shift for each treatment, based on the fitted type-1 model. **e** The phase in which the intervention elicited the maximal phase shift for each intervention. **f** Comparison of period length before and after each treatment (two-sample two-sided Student's $t$ test between the period-change (After–Before) of the UT control and the period-change in the treated group; $n$ cells per condition indicated in brackets. Central circle: median; box: interquartile range (IQR); whiskers: extend to the maximal/minimal values within the range of ±1.5 IQR; circles: outliers). Source data are provided as a Source Data file and in Supplementary Dataset 2.

on the clock. Importantly, the time-dependent effect of signals on the clock is one of the foundational principles of chronobiology as already described and established more than 50 years ago[14]. These pioneering studies have highlighted the importance of PTCs, however, given their laborious nature, experimentalists often refrain from generating full PTCs. Consequently, our

knowledge regarding the effect and potency of different signaling pathways on the clock is rather limited.

It is widely accepted that clocks tick independently in individual cultured cells such as NIH-3T3, and gradually dephase from each other[18,21,22]. Consequently, as above-shown, they exhibit a wide phase representation after few days in culture, which enables

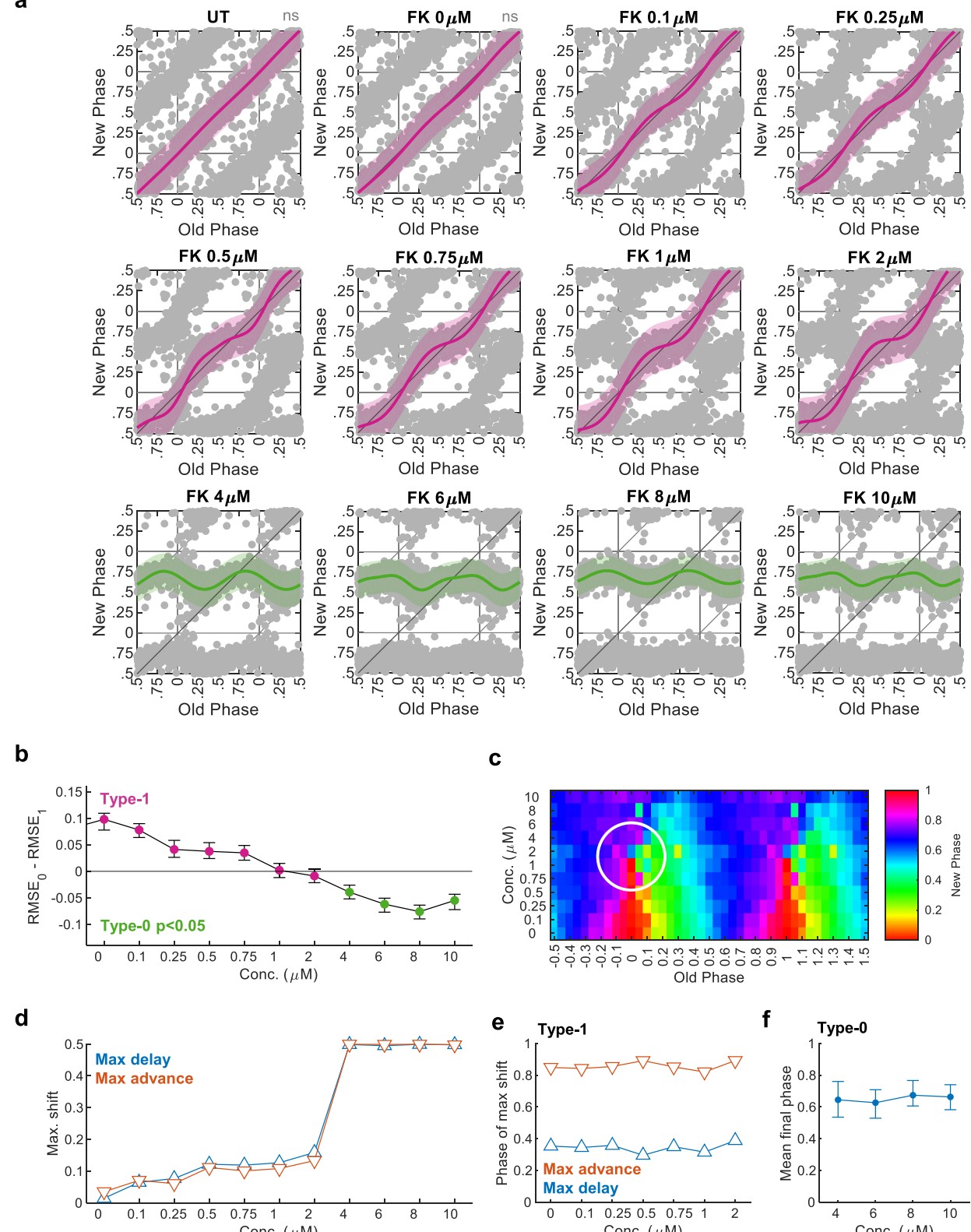

the reconstruction of a PTC by tracking individual cells upon a single intervention. Previous studies have used a similar principle in cyanobacteria[31,38] and plants[39]. The cyanobacteria studies produced the mixed-phase population artificially, by mixing sub-populations which were pre-synchronized to different phases. In the plant study, whole plants were used rather than single cells. Circa-SCOPE employs this principle for the reconstruction of

high-resolution PTCs in mammalian cells and is therefore expected to fill the gap in our knowledge regarding the diversity of circadian clock inputs.

In addition, Circa-SCOPE carries several major advantages over the current methodology for reconstructing PTCs. First, the method provides comprehensive high-resolution PTCs, together with their statistical and topological characterization. Second, it is

**Fig. 4 Three-dimensional high-resolution PTC for Forskolin. a** PTCs from cultures treated with different doses of Forskolin (FK), (UT: untreated, FK 0 μM: DMSO only). (Pink: type-1 resetting model ±95% confidence interval, green: type-0 resetting model ±95% confidence interval, ns nonsignificant ($p > 0.05$) in bootstrapping test for response, $n = 640, 589, 743, 732, 744, 699, 686, 900, 677, 957, 768, 727$ cells per concentration, respectively). **b** Model selection for each concentration. Dots represent the observed $RMSE_0$–$RMSE_1$, error bars represent the 95% confidence interval from bootstrapping test, and color indicates whether the test was significant for type-0 (one-sided). **c** Phase plane representation of the results in (**a**). The $x$-axis represents the Old Phase, and the color represents the New Phase, both are averaged in 0.05 bins. The $y$-axis corresponds to the FK concentration, such that each row corresponds to each PTC in (**a**). The white circle marks the surroundings of the singularity point. **d** Maximal phase shift for each FK concentration, based on the best-fitted model. **e** Phase of maximal phase shift for each concentration in the type-1 PTC range. **f** Mean final phase for each concentration in the type-0 PTC range, based on the type-0 fitted model (error bars represent the range of final phases from the fit). Source data are provided as a Source Data file and in Supplementary Dataset 3.

fully automated, from data acquisition to analysis. Third, it is relatively high throughput, as its current form can measure up to 36 conditions simultaneously (and can be readily upscaled). Finally, it offers a more efficient resource usage (e.g., cells, media, and the tested compounds) that are often limiting, especially in contexts of custom-purified compounds or of large-scale library screens. A simple calculation shows that a single well in Circa-SCOPE that provides a 1 h resolution PTC would be the equivalent of 24-wells using standard analysis (excluding biological replicates).

In this study, we were able to characterize a diverse cohort of resetting agents. For each compound, we could determine not only whether it can reset the clock, but also to compare its relative strength and its PTC's topological characteristics (Summarized in Supplementary Table 1). The qualitative and quantitative differences between the different agents likely reflect molecular differences in their mode of action. For example, cobalt chloride, a hypoxia mimetic, likely signals the clock through HIF1α and its interaction with BMAL1[29,30,40]. PMA, a Protein Kinase C activator, might function via induction of *Per* expression[27], while lithium might act by inhibition of GSK3b, which phosphorylates clock proteins like PER2 (ref. [41]). In this conjuncture, Circa-SCOPE can serve as a powerful tool to experimentally map input pathways to the clock. Pharmacological and genetic loss-of-function experiments for signaling checkpoints and clock components are expected to shed light on the signaling pathways that reset the clocks and map their interaction with the clock at the molecular level. In this regard, our GR/PR antagonist experiments together with serum samples serve as a proof of concept.

In addition, the method enables the systematic study of the combinatorial effect of different zeitgebers on the clock. Basically, by applying different signals together, in various concentration ratios, one can ask whether and how their interaction affects the phase of the clock. As the circadian clock integrates input from a wide variety of signaling pathways to compute a phase[42–45], this type of experiment is expected to provide valuable insights regarding signal integration by a nonlinear genetic network.

There is a fast-growing interest in translational chronobiology, spanning different health and well-being arenas[46,47]. One prominent example is chronotherapy, namely tailoring the time of administration of certain drugs to maximize effectiveness and minimize side effects[48,49]. In this respect, Circa-SCOPE can be utile in the chronobiologist's toolbox. For instance, as a drug screening tool, either for clock-resetting drugs against jetlag, social jetlag, and sleep disturbances, or for characterizing the effect of drugs on the clock as side effects. It is likely that certain compounds might not reset the clocks by themselves, but do affect the resetting characteristics of other signals such as blood-borne compounds. Our compound screens highlight the potential of Circa-SCOPE in this regard.

Aside from the many advantages Circa-SCOPE offers, the method has several limitations. As an in vitro assay, it obviously does not fully recapitulate in vivo resetting and does not take into consideration the in vivo pharmacokinetics and pharmacodynamics of the different compounds

(e.g., steroids). Also, the use of serum which is highly variable between different batches and individuals may carry some confounding effects. Circa-SCOPE currently relies on a single NIH-3T3-based reporter cell line, which exhibits relatively low amplitude oscillations, and thus imposes both biological limitations and technical challenges. The future use of different circadian reporters alongside various cell types including primary cells from human individuals is expected to be of great value.

The data can be analyzed in many ways at multiple stages from fit adjustments, detrending, and phase and rhythmicity analyses. It is noteworthy that our approach is based on filtering out cells that do not present reasonably stationary oscillations within a 3-days timeframe. This point is critical for the reconstruction of PTCs, as the old and new phases and periods must be well-defined. However, this type of analysis can miss dynamic changes that might occur within this time window. In this conjuncture, the raw data provided herein can be reanalyzed in the future using different methods[50–52] that better capture the detailed dynamics of the circadian signal in response to perturbations, and are expected to yield interesting insights. Importantly, the computational pipeline provided herein, based on our different analyses, reliably provides PTCs with topologies characteristics that follow the classical distinction of type 0 or 1.

It is conceivable that various pathological conditions (e.g., diabetes, metabolic syndrome) might not only interfere directly with the molecular clockwork[53,54] but also impair signaling pathways that participate in clock resetting. Hence, the poor rhythmicity that is often observed in various pathologies might stem from disruption of either the core oscillator, input mechanisms to the clock, or both. Valuable information in this conjunction can be obtained by testing the resetting properties of serums from individuals using Circa-SCOPE, for example comparing the resetting properties of serum derived from healthy individuals vs. individuals with different pathologies.

In summary, Circa-SCOPE provides a significant advancement in our ability to study circadian clock resetting compared to currently available methods and therefore opens the door for expanding our knowledge on circadian clock properties in both basic and translational research.

## Methods

**Establishing a dual reporter cell line**. The dual-reporter cell line was established on the basis of the NIH-3T3 Rev-VNP-1 cells[7]. These cells express Venus fluorescence protein fused to NLS (nuclear localization sequence) and PEST domains, under the promoter of the clock gene *Reverbα*. The cells were infected with pLenti6-H2B-mCherry nuclear fluorescence reporter vector (Addgene plasmid # 89766)[17], and then treated with Blasticidin (2.5 μg/ml) for 1 week to select for stably expressing cells.

**Cell culture and live imaging**. For general maintenance, cells were grown in standard high glucose DMEM (Biological Industries) supplemented with 10% FBS (Gibco), 1 mM Glutamine (Biological Industries), 100 units/mL penicillin, and 100 mg/mL streptomycin (Biological Industries).

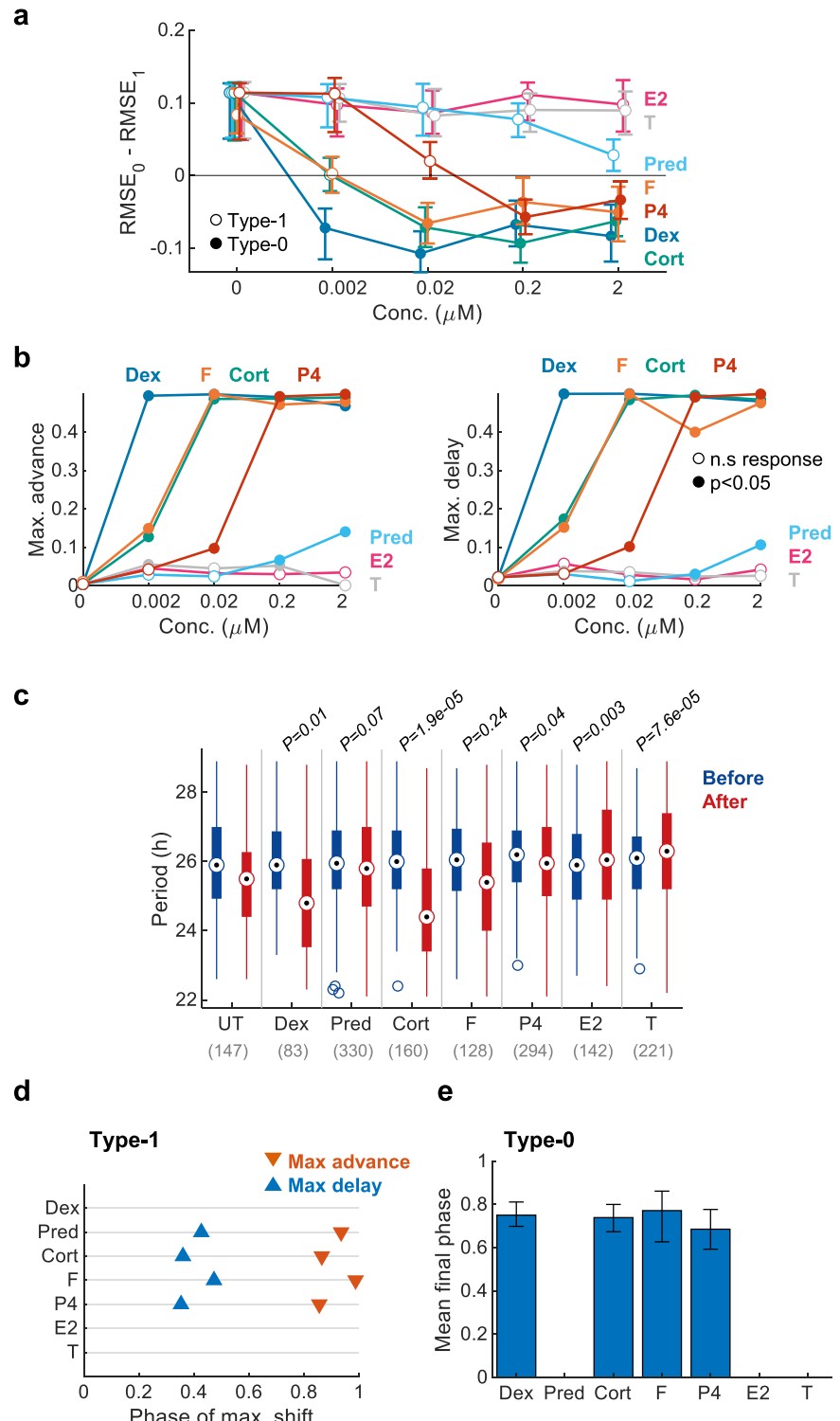

For each round of Circa-SCOPE, cells were split into 24-well plates at a concentration of 12,500 cells per well, in the above-described media. Cell counting was performed using a TC20 automated cell counter (BioRad). After 24 h, the medium was replaced with FluoroBright DMEM (Gibco) with Glutamine, Penicillin-Streptavidin, and 1% FBS, in a volume of 2 ml per well. The FluoroBright medium decreases background fluorescence, while the reduced serum levels reduce cell proliferation and increase the population stability. After 3 more days, cells were entered into the IncuCyte ZOOM (Essen Bioscience) microscope, inside a humidified, 37 °C, 5% $CO_2$ incubator.

Time-lapse microscopy imaging was performed with a 10× magnification lens, in three channels: Phase-contrast, Green (excitation: 440–480 nm, emission:

504–544 nm), and Red (excitation: 565–605 nm, emission: 625–705 nm). The green channel captures the *Rev-VNP* reporter, while the red channel captures the *H2B-mCherry* reporter. Phase-contrast images were only used for general inspection of the population status and were not used further in the analysis. Images were taken every 60 min for a total duration of 9 days, in 16 fields-of-views per well. After 4.5 days since the beginning of imaging, plates were taken out of the incubator, treated with the agent of choice by adding it on top of the existing media, and placed back in the microscope. The recording was then continued for another 4.5 days.

Specific compounds that were used for resetting experiments are detailed in Supplementary Table 2.

**Fig. 5 Steroid drugs and hormones screen using Circa-SCOPE.** Cell cultures were treated with 7 different steroids: Dexamethasone (Dex; $n$ = 147, 169, 202, 157, 83 per concentration), Hydrocortisone (F; $n$ = 179, 210, 187, 112, 128 per concentration), Corticosterone (CORT; $n$ = 147, 295, 209, 193, 160 per concentration), Progesterone (P4; $n$ = 147, 152, 290, 299, 294 per concentration), Prednisone (Pred; $n$ = 147, 214, 174, 346, 330 per concentration), β-Estradiol (E2; $n$ = 147, 148, 207, 270, 142 per concentration), and Testosterone (T; $n$ = 147, 311, 202, 238, 221 per concentration). **a** Model selection for each drug across the tested concentrations. Dots represent the observed $RMSE_0 - RMSE_1$, error bars represent the 95% confidence interval from bootstrapping test. Filled dots stand for significant type-0 fit according to bootstrapping test. **b** Maximal phase advance (left) and maximal phase delay (right) for each compound across the tested concentrations. Empty dots represent non-significant phase shifts based on bootstrapping test. (**c**) Comparison of period length before and after treatment with 2 μM of each drug. (Two-sample two-sided Student's $t$ test between the period-change (After–Before) of the UT control and the period-change in the treated group; $n$ cells per condition indicated in brackets. Central circle: median; box: interquartile range (IQR); whiskers: extend to the maximal/minimal values within the range of ±1.5 IQR; circles: outliers). **d** Phase of maximal shift for the maximal concentration with significant type-1 resetting with each compound. Data for compounds without significant type-1 response in any concentration are omitted. **e** Mean final phase for the maximal concentration with significant type-0 resetting with each compound (error bars represent the range of final phases from the fit). Data for compounds without significant type-0 response in any concentration are omitted. Source data are provided as a Source Data file and in Supplementary Dataset 4.

The use of commercial human serum was approved by Weizmann's Institutional Research Board. Rat serum was obtained in conformity with the Weizmann Institute Animal Care and Use Committee guidelines and kindly provided by Avigdor Schertz's lab.

**Image analysis**. Image analysis was performed using the CellProfiler platform (version 3.9.1, Broad Institute)[55], with a customized pipeline.

First, red and green channels illumination inconsistencies were calculated with the "splines" method and then subtracted from the original images. For the red-channel images, feature enhancement was performed, to increase the contrasts of speckle-shaped features (representing nuclei) of typical width of 25 pixels. Next, the corrected and enhanced red-channel images were used for primary object identification, which segments the nuclei. The object identification was performed with the following specifications: minimal diameter = 9 and maximal diameter = 35. Larger objects and objects touching the borders of the images were excluded. Global thresholding with robust background method was used, with 0.05 lower and upper limits for outliers, on medians with 5 standard deviations threshold, and smoothing scale of 0.8. Clamped objects were distinguished by the intensity and dividing lines between them were drawn by shape. The size of the smoothing filter for declumping, and the minimum distance allowed between local maxima, were automatically calculated. Holes in objects were filled after declumping.

Next, to reduce the effect of fluctuations in background intensity in the green channel on our measurements, the average green intensity outside nuclei was subtracted from the green images. The background-subtracted green intensities inside the nuclei were then measured.

The identified nuclei were tracked using the "Follow Neighbors" method with the following specifications: maximum distance of 50 pixels, average object diameter of 20 pixels, cost of cells to the empty matching of 100, the weight of area difference in function matching cost of 100. Objects were filtered by lifetime with a minimum lifetime of 24 h. The "Follow Neighbors" method also overcomes the problem of frame drifting, which is challenging especially when a plate is taken out for treatment and then replaced back into the microscope.

All measurements of object intensities and tracks were exported to csv file. In a typical experiment, raw images from a single well reach a total size of ≈10 GB, and the CellProfiler result files are ≈2 GB. Hence, the data size for a 36-well experiment is ≈450 GB.

The CellProfiler pipeline was run on the Weizmann's institutional computation cluster WEXAC, using the CellProfiler batch analysis procedure.

**Data processing and PTC reconstruction**. Post-processing and further analysis of the pipeline results were performed using a MATLAB (MathWorks, version R2020b) script.

Analysis was performed on a per well basis. After loading all the data to MATLAB, each tracked nucleus was given a unique numerical identifier that distinguished it for the entire tracking duration. Then, only tracks that span the entire experiment were selected for subsequent analysis. Their green corrected and background-subtracted intensities were detrended with Z-scoring: a trend of running average with a 48 h window was calculated, and then subtracted from the data and divided by the standard deviation of that track.

Subsequently, the detrended intensities were used for the rhythmicity analysis. Each track was fitted with a cosine, using the "harmfit" function[56] in two windows: before the treatment (between time-points 24–96 h) and after the treatment (120–196 h). The fit was performed on a range of period lengths between 20 and 32 h with 0.1 h increments. The best fit was selected by maximizing the $R^2$ (calculated with the "rsquared" function[57]), and its phase, period, and amplitude parameters were extracted. To calculate the phase difference upon the treatment, the phases were adjusted to period differences using the TIPA method[23]. In short, the relative phase of the before fit in the time of cue was calculated and then used to

create a virtual extension of the fit starting from this phase onward with a period length equal to the one of the after fit. This new trace, termed the "null" curve, is used instead of the before fit in the construction of the PTC. In this way, the old and new phases are derived from cosines with the same period length. All phases were divided by their period lengths, to get scaled phases in the range of [0,1].

For PTC construction the fit-driven phases were filtered for the goodness of fit of $R^2 > 0.5$, and for period length between 22 and 29 h.

**Circular statistics**. All circular statistics (circular means, standard deviations and correlations, and the $p$-values thereof) were calculated with CircStat 2012 (ref. [58]) in MATLAB.

**Rhythmicity analysis evaluation**. Evaluation of our rhythmicity filter was conducted against three alternative rhythmicity analysis algorithms. ARSER and JTK were performed via the MetaCycle package (v1.2.0)[59], and RAIN was conducted using the RAIN package (v1.24.0)[60], both in R (4.0.3).

**PTC statistical analysis**. PTCs were characterized based on a Fourier model selection, following bootstrapping-based hypothesis testing. The Fourier model was fitted using the MATLAB $fit$ function, with "fourier2" model. The model was fitted twice, for either type-1 or type-0 resetting. For type-1, the data were transformed to PRC format (i.e., the $y$-axis represents the extent of phase-shift rather than the final phase) and duplicated along the $x$-axis prior to fitting. Because the Fourier model oscillates around a horizontal line, this transformation ensures the average slope of 1 on PTC, as required from a type-1 curve. For type-0, the data remained in the PTC format and was also duplicated along the $x$-axis.

To test whether any phase-response is present in a certain group, we bootstrapped from control (untreated) group samples in the size of the test group for 10,000 times. In each iteration, a Fourier model was fitted, and the maximal shift was retrieved. The null hypothesis (namely, no phase-shift) was rejected if the resampled maximal shift was higher than the observed maximal shift in less than 5% of the time.

To select between type-1 and type-0 models, the root-mean-square deviation (RMSE) was calculated for each fit (retrieved from the $fit$ function). The model which fits better should exhibit a lower RMSE than its competitor. To test whether the selected RMSE is significantly lower than the other, we bootstrapped from the tested group 10,000 times. In each iteration, both models were fitted to the resampled data, and RMSE was calculated for them. The type-1 RMSE should be larger than the type-0 RMSE in over 95% of the time in order to reject the null hypothesis of type-1 resetting.

To compare between two groups, the null hypothesis of no difference between the conditions in a certain parameter was tested. Data was bootstrapped 10,000 times, each time two independent samples are drawn, one from each group, with corresponding sizes. In each iteration, the parameter is calculated for each sample and the difference between the parameters is retrieved. The null hypothesis is rejected when 0 is outside of the 95% confidence interval of the bootstrapping-based distribution of differences.

**Conventional PTC construction with NIH-3T3 Bmal1-Luc-1 cells**. NIH-3T3 Bmal1-luc-1 cells[7] were seeded at a concentration of 62,500 cells per 3.5 cm plate. After 24 h, culture media was replaced with the same recording media described above for the microscopy experiments (i.e., FluoroBright-based). After 6 days, media was replaced with fresh media containing 100 mM of D-luciferin (Promega), and plates were subsequently placed in the LumiCycle32 (Actimetrics) for live bioluminescence recording at 37 °C and 5% $CO_2$. After 2 days of recording, dex-amethasone was added dropwise to the media to a final concentration of 100 nM, at 4 h intervals over 24 h. The recoding was continued for three additional days. The PTC was reconstructed based on the time of the peak on the second day following the treatment.

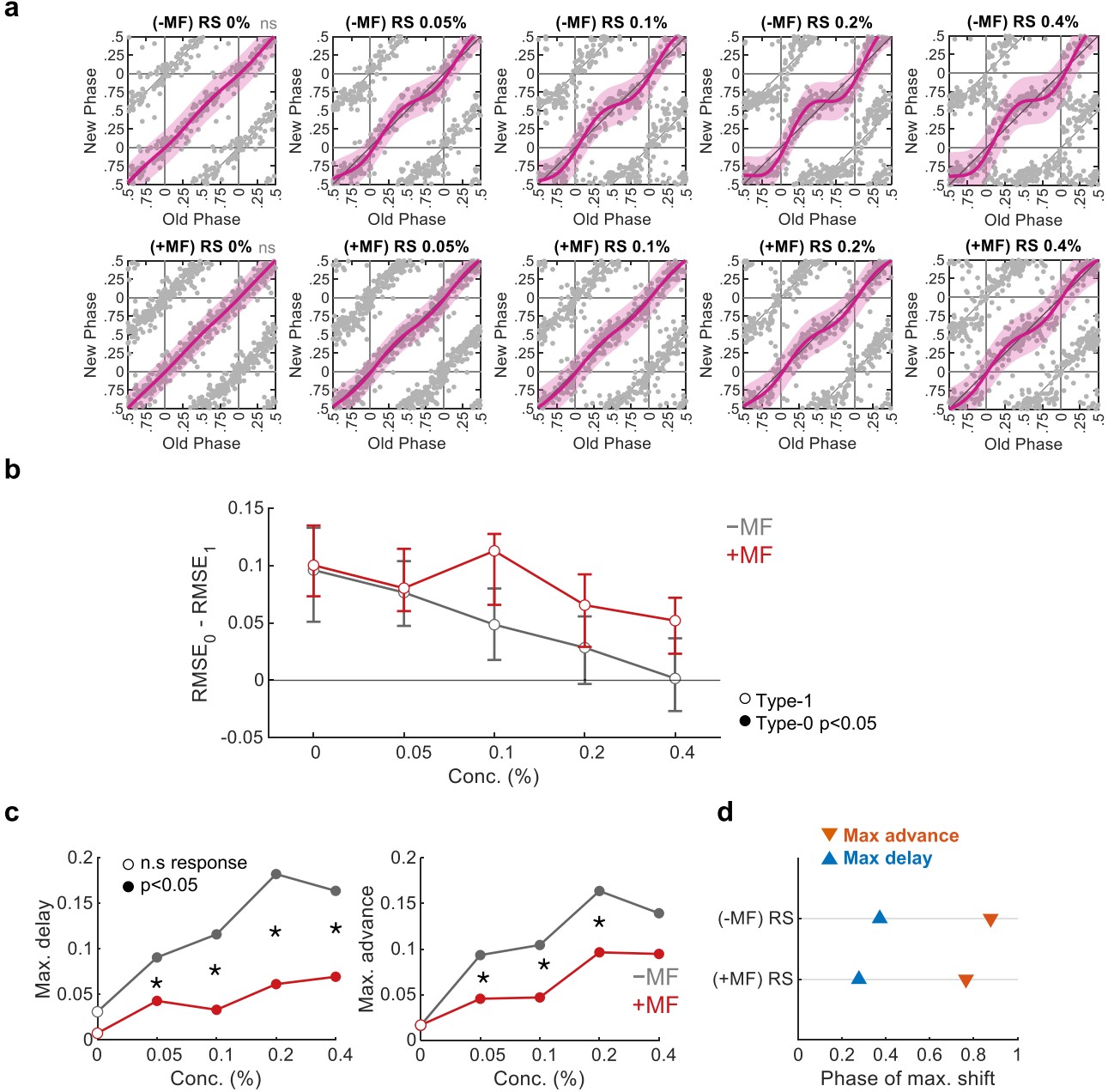

**Fig. 6 Inhibition of the glucocorticoid and progesterone receptor affects serum-induced resetting. a** PTCs of cultures in response to different concentrations of rat serum (RS), with or without pretreatment of 1.25 μM mifepristone (MF). (Pink: type-1 resetting model ±95% confidence interval, ns not significant ($p > 0.05$) in bootstrapping test for response, $n = 102, 126, 155, 161, 162, 212, 235, 166, 143, 173$ cells per condition, respectively). **b** Model selection for each treatment across the tested concentrations. Dots represent the observed $RMSE_0$–$RMSE_1$, error bars represent the 95% confidence interval from bootstrapping test. Filled dots stand for significant type-0 fit according to bootstrapping test. **c** Maximal phase advance (left) and maximal phase delay (right) for each treatment across the tested concentrations. Empty dots represent nonsignificant phase shifts based on bootstrapping test. (*$p < 0.05$, two-sided bootstrapping test for the difference in maximal shift between two conditions). **d** Phase of maximal shift induced by 0.4% RS with and without MF. Source data are provided as a Source Data file and in Supplementary Dataset 6.

## Data availability

Demo image sets (the raw data of Fig. 2 experiment) that can serve as input for the CellProfiler pipeline are available in Zenodo under doi.org/10.5281/zenodo.5139326. Demo CellProfiler outputs, which can serve as input for the MATLAB code, are available in Zenodo under doi.org/10.5281/zenodo.4588349. Due to the large size of the datasets and lack of adequate public depository, the rest of the raw images generated in this study are available upon request from the authors. The processed data (raw individual cell traces) generated in this study have been deposited in Zenodo under doi.org/10.5281/zenodo.5139746. The Circa-SCOPE results generated in this study are provided in Supplementary Data 1–7. Supplementary_Data_1.xlsx is associated with Fig. 2, and

Supplementary Figs. 3 and 4, Supplementary_Data_2.xlsx is associated with Fig. 3 and Supplementary Fig. 5. Supplementary_Data_3.xlsx is associated with Fig. 4 and Supplementary Fig. 6. Supplementary_Data_4.xlsx is associated with Fig. 5 and Supplementary Fig. 7. Supplementary_Data_5.xlsx is associated with Supplementary Fig. 8. Supplementary_Data_6.xlsx is associated with Fig. 6. Supplementary_Data_7.xlsx is associated with Supplementary Fig. 9. Source data are provided with this paper.

## Code availability

The CellProfiler custom pipeline and the MATLAB code for post-analysis are available through GitHub[61].

# ARTICLE

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

## Acknowledgements
We are grateful to all the members of the Asher lab for their comments on the paper. We wish to thank Uriel Rosen (WIS IT unit) and Jonathan Sobel for their assistance in establishing the CellProfiler pipeline on the institutional cluster. G.A. is supported by a grant from the European Research Council (ERC-2017 CIRCOMMUNICATION 770869), Abisch Frenkel Foundation for the Promotion of Life Sciences, Adelis Foundation, Susan and Michael Stern.

## Author contributions
Conceptualization: G.M. and G.A.; Investigation: G.M., D.A., R.A. and M.G.; Visualization: G.M.; Data curation: G.M.; Software: G.M.; Funding: G.A.; Writing—review & editing: G.M. and G.A.

## Competing interests
A patent application has been submitted by G.M., R.A., and G.A. to the Israeli Patent Office and is currently under consideration. The remaining authors declare no competing interests.
