## [Peer Review File · Nature Communications]

Reviewer comments, initial- -

Reviewer #1 (Remarks to the Author):

In this manuscript, the authors developed a method to rapidly characterize the phase response curve of cell culture models in vitro. This is interesting because in an unsynchronized culture, the initial phases of each cell are desynchronized, and therefore represent many/all potential phases. So, a comprehensive PRC can be done at a single time point. On the plus side, this overcomes a substantial sampling problem and greatly accelerates the conduct of these type of experiments. The authors go on to show the utility of this method looking at many perturbations of the NIH3T3 model, including chemical resetting agents and serum. They find differences, for example, in how cobalt chloride, a hypoxia mimetic, shifts clocks from immediate early gene triggers such as forskolin or serum. The manuscript has many strengths. The paper is clearly written and organized, the central hypothesis is elegant and simple, and the results are compelling. Tackling all initial phases at once is a major, hard to overstate advantage of this method. The manuscript also has some weaknesses. the in vitro models of the clock are relatively low amplitude, change period during the experiment (which the authors address), and may or may not reflect what's going on with conventional behavioral resetting by light resetting of the SCN clock. The paper would benefit from a discussion at the end of this and any other potential weaknesses to give it a bit more balance. Overall, I think this paper will be of high interest, principally to people in the field. Some points for the authors to address.

1. Does the classical type 0 and type 1 resetting models hold?
2. Cobalt chloride is likely stimulating the HIFs to interact with Bmal1 and reset the clock in that manner, whereas PMA is likely working through immediate early gene up regulation of the Per genes. These aspects are not discussed.
3. The experiments with serum are complicated to interpret, in part because when you buy serum, you never get the same product twice. It is a biological material and constantly changing. This caveat should be noted.
4. Interpretation of steroid PRCs is likely complicated by the lack of knowledge of how these compounds behave ex vivo. The PK/PD of these steroids is far more characterized in vivo than ex vivo.

Reviewer #2 (Remarks to the Author):

Manella et al. propose a methodology to generate PTCs from individual mammalian cells, named Circadian Single-Cell Oscillators PTC Extraction (Circa-SCOPE). Employing this protocol, the authors calculated the resetting capacity of various steroids and their relative strength, with their statistical and topological characterization.

General comments

The authors have identified a very exciting and challenging problem of deriving PTCs from individual cell recordings of mammalian cells.

The efficient process of obtaining a PTC from single-cell traces relies on three main components:

1. Having a desynchronized population of oscillators at the time of the perturbation.
2. The capability to track continuously individual cells for prolonged periods of time.
3. A precise phase estimation before and after the treatment.

The 1st step is a direct consequence of working with uncoupled or weakly coupled single-cell circadian oscillators with their characteristic period heterogeneity. This is not part of an experimental design from the authors but practically unavoidable and frequently observed both in the NIH-3T3 and U2OS cell lines using this Rev-VNP reporter. The 2nd step is routinely achieved in multiple signaling-dynamics studies relying on single cell tracking (Spencer et al., 2013; Reyes et al., 2018) and there are multiple robust and well established dedicated software for this purpose. Unfortunately, the third and potentially innovative contribution of this manuscript, i.e., to obtain a phase estimation from single cell recordings contains several data analysis deficiencies that could be improved. Consequently, the current version of the manuscript fails to provide an accurate

solution for obtaining single-cell derived PTCs. On the other hand, I believe that the authors have performed a wide and clever set of experiments and no additional experiments are needed. The data analysis issues can be resolved by implementing appropriate single-cell statistical and time series analysis methods.

Phase extraction.

A PTC summarizes the phase-shifting effects that perturbations induce on an oscillator. The most critical step when generating PTCs is a robust phase-shifting estimation. To calculate a phase shift is necessary to precisely determine the phase before and after a given perturbation. In systems that oscillate with a sinusoidal stationary waveform, a sinusoidal fit is a powerful and simple technique to extract period, amplitude, and phase. On the other hand, single-cell circadian recordings like the ones analyzed here are characterized by a very noisy amplitude, non-sinusoidal waveforms, and fluctuating period (Bieler et al., 2014; Feillet et al., 2014). Utilizing a single sinusoidal fitting function, what the authors call "cosine-model" is likely an inaccurate method for these single-cell circadian signals because it wrongly assumes stationarity and waveform conservation. The authors should consider implementing a data analysis strategy better suited to their data. Otherwise, their results are potentially inaccurate and the conclusions misleading. Fortunately, there are multiple data analysis strategies already available that are exactly designed for this purpose. (Price et al., 2008; Leise et al., 2012; Leise, 2013; Bieler et al., 2014; Mönke et al., 2020).

In addition, the authors provide supplementary datasets of their own parameter estimations but, to allow an independent analysis, the authors could provide all raw time series data from their tracked cells. This supplementary raw data should also contain the excluded cells (those who fail outside their period cut-off and those with $R^2 < 0.5$) and should be non-smoothed, non-detrended, non-interpolated data. Without these datasets it is difficult to independently judge the accuracy of their phase estimation.

Rhythmicity

The authors utilize a metric for rhythmicity based on their single-sinusoidal fit function. This is not a robust state-of-the-art metric of rhythmicity. Furthermore, the authors utilize the same fitting strategy as a dual approach of filtering-out their dataset (removing cells with $R^2 < 0.5$) and later for the phase estimation. The authors should utilize multiple and independent methodologies to filter their data by rhythmicity and multiple and independent methodologies for the phase estimation. There are diverse available tools in the circadian literature specifically designed to determine the rhythmicity of circadian signals (e.g. Moore et al., 2014; Thaben and Westermarck, 2014; Cenek et al., 2020). This manuscript will strongly benefit from an accurate and state-of-the-art metric of rhythmicity. This is particularly important when using rhythmicity as a method to filter out traces that will not be used to derive the final PTCs.

Detrending

The authors use a moving average (MA) with a 24h time window as detrending strategy. Moving average filters are known to introduce spectral artifacts. The authors could avoid detrending their signal or implement some of the multiple alternative methods to detrend signals that are less prone to spectral artifacts. Using a MA filter is specially concerning when using it with a window size that coincides with the period of the signal of interest (~ 24 h). The authors should utilize a filter method less prone to artifacts and/or utilize a window size far from the period of the signal of interest and/or provide the spectral analysis before and after their detrending approach to allow an analysis of how their detrending approach potentially perturbs (or not) the power of the circa 24 hours oscillatory components.

PTC reconstruction

1. This manuscript aims to determine the phase shifting effect to a given strength of a perturbation, which then the authors argue that it can be used to understand the dynamics of dose-dependency. In this context it is critical to quantitatively characterize if the phase shift estimated by the authors is only a part of a transient phase shift or a stable phase shift. This could be estimated by obtaining the instantaneous phase post treatment and plot the instantaneous phase-shift as a function of time. If the phase-shift reaches a plateau, then it can be considered a stable phase shift. Otherwise, it is just a partial transient phase estimation limited by the observation time-window.

2. Given a common perturbation, an oscillator with small amplitude will suffer stronger phase shifts than an oscillator with a bigger amplitude. In other words, the phase shifting power of a perturbation is strongly affected by the amplitude of the perturbed oscillator. The amplitude of these single-cell circadian oscillators is known to strongly fluctuate in time and to vary from peak to peak. Therefore, to determine the phase-shifting effect of the strength of a perturbation, the authors should estimate the relative amplitude at the time of treatment and not the averaged amplitude before treatment. Moreover, the amplitude of different single-cell oscillators is known to be extremely variable, with some oscillators having $\sim 3X$ higher amplitude than others. A perturbation normalized to the different amplitudes of the single-cells and normalized to the instantaneous amplitude within a single oscillator is critical for the topological analysis of the PTC and the dynamics of dose-dependency. From the mathematical perspective a phase-shifting perturbation drives the phase across multiple isochrons, and the final phase-shift is determined by the difference between the starting isochron and the final isochron. The number of isochrones crossed (final phase-shift) is a direct consequence of the relationship between the ratio of the oscillator amplitude over the perturbation strength (dose), the direction of the perturbation and the particular isochron spatial structure. In oscillators with strong amplitude fluctuations, the amplitude estimation should be based on an instantaneous amplitude estimation (Guckenheimer, 1975; Gunawan and Doyle, 2006).

Single cell tracking

The authors record their cells using an Incucyte system seeding their cells in a 24-well plate and imaging with long focal distance 10X objective. With this plate (not the image-lock 96-well plate), the Incucyte system is not capable of perfect stage positioning from one time-point to the next time-point, which means that the cells are not in the same location along the consecutive recordings. This is a known issue, as the authors also acknowledge, that strongly limits single-cell tracking with the Incucyte data. Therefore, the vast majority of Incucyte users do not perform single-cell tracking with these images and only work with population-based analysis. The authors have resolved this challenging problem with a custom-made Cellprofiler pipeline. I consider this a very exciting step and one of the main challenges for readers to be able to implement the method presented in this manuscript. Here, the authors could provide their Cellprofiler pipeline for testing together with a set of movies examples where the single-cell tracking performs as expected so the reader would be able to reproduce the presented results.

Additional comments

1. Please add page numbers and line numbers to facilitate the review process.
2. The authors claim that their method can estimate phase and period. To determine phase the period calculation is mandatory. In other words, these are not independent capabilities of their approach but a single one.
3. High-throughput: One of the main advantages that the authors claim is that Circa-SCOPE provides a "high-throughput" method, which is in contrast with the classical approach of generating PTCs. The "high-throughput" part lies partially on the fact that these cells are out-of-sync at the time of the perturbation. This is a known observation and expected result of having an uncoupled heterogeneous population of single-cell oscillators and not a specific experimental design from the authors.
4. The authors argue that their method provides a high-resolution PTC. This statement remains unclear to me. If this is related to the 30-min resolution, then the authors should clarify how they obtain a 30min resolution out of Incucyte recordings performed each 60min. Is this high-resolution is a consequence of fitting and/or interpolation?

References

- Bieler, J, Cannavo, R, Gustafson, K, Gobet, C, Gatfield, D, and Naef, F (2014). Robust synchronization of coupled circadian and cell cycle oscillators in single mammalian cells. *Mol Syst Biol* 10, 739.
- Cenek, L, Klindziuk, L, Lopez, C, McCartney, E, Burgos, BM, Tir, S, Harrington, ME, and Leise, TL (2020). CIRCADA: Shiny Apps for Exploration of Experimental and Synthetic Circadian Time Series with an Educational Emphasis. *J Biol Rhythms* 35, 214–222.
- Feillet, C et al. (2014). Phase locking and multiple oscillating attractors for the coupled mammalian

clock and cell cycle. *Proc Natl Acad Sci U S A* 111, 9828–9833.

Guckenheimer, J (1975). Isochrons and phaseless sets. *J Math Biol* 1, 259–273.

Gunawan, R, and Doyle, FJ (2006). Isochron-based phase response analysis of circadian rhythms. *Biophys J* 91, 2131–2141.

Leise, TL (2013). Wavelet analysis of circadian and ultradian behavioral rhythms. *J Circadian Rhythms* 11, 5.

Leise, TL, Wang, CW, Gitis, PJ, and Welsh, DK (2012). Persistent Cell-Autonomous Circadian Oscillations in Fibroblasts Revealed by Six-Week Single-Cell Imaging of PER2::LUC Bioluminescence. *PLoS One* 7, 1–10.

Mönke, G, Sorgenfrei, F, Schmal, C, and Granada, A (2020). Optimal time frequency analysis for biological data - pyBOAT. *BioRxiv* 179, 985–986.

Moore, A, Zielinski, T, and Millar, AJ (2014). Online Period Estimation and Determination of Rhythmicity in Circadian Data, Using the BioDare Data Infrastructure. *Methods Mol Biol* 1158, 13–44.

Price, TS, Baggs, JE, Curtis, AM, FitzGerald, GA, and Hogenesch, JB (2008). WAVECLOCK: wavelet analysis of circadian oscillation. *Bioinformatics* 24, 2794–2795.

Reyes, J, Chen, JY, Stewart-Ornstein, J, Karhohs, KW, Mock, CS, and Lahav, G (2018). Fluctuations in p53 Signaling Allow Escape from Cell-Cycle Arrest. *Mol Cell* 71, 581-591.e5.

Spencer, SL, Cappell, SD, Tsai, FC, Overton, KW, Wang, CL, and Meyer, T (2013). XThe proliferation-quiescence decision is controlled by a bifurcation in CDK2 activity at mitotic exit. *Cell* 155, 369–383.

Thaben, PF, and Westermark, PO (2014). Detecting Rhythms in Time Series with RAIN. *J Biol Rhythms* 29, 391–400.

We would like to thank the reviewers for their fair and constructive comments. We made considerable efforts to address virtually all of their suggestions by reanalyzing the data as well as incorporating some textual changes as detailed below.

We followed the suggestions of referee #2 to apply different computational pipelines and reanalyzed the entire data. We replaced the original figures with the new analyses. Importantly, the results, presented in the revised manuscript, are very similar to the one provided in the original manuscript, and did not affect our conclusions. In addition, we provide comparisons of different computational approaches in new Supplementary Fig. 1, and 3 that support the validity of our analyses.

We also deposited the entire raw data in to Zenodo, so they are readily accessible for other labs to extract and analyze according to their research interest.

Response to reviewers' comments:

Reviewer #1

In this manuscript, the authors developed a method to rapidly characterize the phase response curve of cell culture models in vitro. This is interesting because in an unsynchronized culture, the initial phases of each cell are desynchronized, and therefore represent many/all potential phases. So, a comprehensive PRC can be done at a single time point. On the plus side, this overcomes a substantial sampling problem and greatly accelerates the conduct of these type of experiments. The authors go on to show the utility of this method looking at many perturbations of the NIH3T3 model, including chemical resetting agents and serum. They find differences, for example, in how cobalt chloride, a hypoxia mimetic, shifts clocks from immediate early gene triggers such as forskolin or serum. The manuscript has many strengths. The paper is clearly written and organized, the central hypothesis is elegant and simple, and the results are compelling. Tackling all initial phases at once is a major, hard to overstate advantage of this method. The manuscript also has some weaknesses. The in vitro models of the clock are relatively low amplitude, change period during the experiment (which the authors address), and may or may not reflect what's going on with conventional behavioral resetting by light resetting of the SCN clock. The paper would benefit from a discussion at the end of this and any other potential weaknesses to give it a bit more balance. Overall, I think this paper will be of high interest, principally to people in the field. Some points for the authors to address.

We are pleased that the reviewer found our piece of high interest. Following the reviewer's suggestions, we included a paragraph detailing the limitations of the new methodology in the discussion section of the revised manuscript.

1. Does the classical type 0 and type 1 resetting models hold?

Overall, the different resetting topologies presented in the study follow the classical distinction between type 0 and type 1. We highlight it in the discussion section.

2. Cobalt chloride is likely stimulating the HIFs to interact with Bmal1 and reset the clock in that manner, whereas PMA is likely working through immediate early gene up regulation of the Per genes. These aspects are not discussed.

The reviewer raises a relevant point, namely the molecular interaction between clock components and different signaling pathways that reset the clock. We refer to some relevant examples in the discussion section of the revised manuscript.

3. The experiments with serum are complicated to interpret, in part because when you buy serum, you never get the same product twice. It is a biological material and constantly changing. This caveat should be noted.

The reviewer is absolutely right, we note it in the revised manuscript in the discussion section.

4. Interpretation of steroid PRCs is likely complicated by the lack of knowledge of how these compounds behave ex vivo. The PK/PD of these steroids is far more characterized in vivo than ex vivo.

We agree with the reviewer that the interpretation of the steroid PRC might be less straightforward in view of the limited knowledge how these compounds act *in vitro* and we discuss this point in the revised manuscript while describing the limitations of our methodology.

Reviewer #2:

The authors have identified a very exciting and challenging problem of deriving PTCs from individual cell recordings of mammalian cells.

The efficient process of obtaining a PTC from single-cell traces relies on three main components:

- 1. Having a desynchronized population of oscillators at the time of the perturbation.*
- 2. The capability to track continuously individual cells for prolonged periods of time.*
- 3. A precise phase estimation before and after the treatment.*

The 1st step is a direct consequence of working with uncoupled or weakly coupled single-cell circadian oscillators with their characteristic period heterogeneity. This is not part of an experimental design from the authors but practically unavoidable and frequently observed both in the NIH-3T3 and U2OS cell lines using this Rev-VNP reporter. The 2nd step is routinely achieved in multiple signaling-dynamics studies relying on single cell tracking (Spencer et al., 2013; Reyes et al., 2018) and there are multiple robust and well established dedicated software for this purpose. Unfortunately, the third and potentially innovative contribution of this manuscript, i.e., to obtain a phase estimation from single cell recordings contains several data analysis deficiencies that could be improved. Consequently, the current version of the manuscript fails to provide an accurate solution for obtaining single-cell derived PTCs. On the other hand, I believe that the authors have performed a wide and clever set of experiments and no additional experiments are needed. The data analysis issues can be resolved by implementing appropriate single-cell statistical and time series analysis methods.

Phase extraction.

A PTC summarizes the phase-shifting effects that perturbations induce on an oscillator. The most critical step when generating PTCs is a robust phase-shifting estimation. To calculate a phase shift is necessary to precisely determine the phase before and after a given perturbation. In systems that oscillate with a sinusoidal stationary waveform, a sinusoidal fit is a powerful and simple technique to extract period, amplitude, and phase. On the other hand, single-cell circadian recordings like the ones analyzed here are characterized by a very noisy amplitude, non-sinusoidal waveforms, and fluctuating period (Bieler et al., 2014; Feillet et al., 2014). Utilizing a single sinusoidal fitting function, what the authors call “cosine-model” is likely an inaccurate method for these single-cell circadian signals because it wrongly assumes stationarity and waveform conservation. The authors should consider implementing a data analysis strategy better suited to their data. Otherwise, their results are potentially inaccurate and the conclusions misleading. Fortunately, there are multiple data analysis strategies already available that are exactly designed for this purpose. (Price et al., 2008; Leise et al., 2012; Leise, 2013; Bieler et al., 2014; Mönke et al., 2020).

We thank the reviewer for the valuable input. We agree that single-cell circadian signal tends to be noisier and that phase estimation is unarguably the most critical step in PTC reconstruction. Yet, we are convinced that in order to reconstruct a meaningful PTC, one has to assume a decent level of stationarity. If the period and phase fluctuate too much, they will mask any deterministic effect of the zeitgeber before it could be even measured. To ensure that we have a reliable phase estimate, we filtered the data specifically for cells which have a significant stationary signal. Furthermore, as our main goal is to reconstruct reproducible PTCs we reason that a priori the assumption of stationarity is essential.

Following the reviewer's suggestion, we tested the wavelet approach as an alternative. However, the resulting instantaneous phase estimate is too affected by local signal variability and misses the overall trend, consequently, it generates noisier PTCs. The other method used in the literature to get more dynamic estimates of phase is peak detection directly from the raw data. We, therefore, compared the fit-based phase estimates to the peaks (local maxima) locations throughout the fitting window. The results are depicted in new Supplementary Fig. 1e-f, and show that there is no systematic bias in our estimates, and that the deviation between the two methods is low (SD less than 1h to each direction). We also included random examples of the data together with their cosine fits, in new Supplementary Fig. 1a, to further support the adequacy of our approach.

As can be appreciated from these analyses, the fit efficiently captures the oscillatory component of the signal, and the signals are quite robust. Finally, it is important to note that while it is true that our method might filter-out cells which are not "stationary" enough, the results correspond well to the entire population signal (i.e., the way tissue-culture PTCs were reconstructed thus far) and are reproducible. We commented on these important points in the revised manuscript.

We agree that that data can be analyzed via many different ways each has its pros and cons and that a more dynamic method has the potential to provide new interesting insight, yet this is beyond the scope of the current study. We certainly hope that the scientific community would benefit from the data we generated and provide herein (see below) for future interesting analyses.

In addition, the authors provide supplementary datasets of their own parameter estimations but, to allow an independent analysis, the authors could provide all raw time series data from their tracked cells. This supplementary raw data should also contain the excluded cells (those who fail outside their period cut-off and those with $R^2 < 0.5$) and should be non-smoothed, non-detrended, non-interpolated data. Without these datasets it is difficult to independently judge the accuracy of their phase estimation.

We fully concur the reviewer that providing the raw untouched data would be valuable for the scientific community for future independent analyses. We uploaded the full raw data to Zenodo: doi:10.5281/zenodo.5139746.

Rhythmicity

The authors utilize a metric for rhythmicity based on their single-sinusoidal fit function. This is not a robust state-of-the-art metric of rhythmicity. Furthermore, the authors utilize the same fitting strategy as a dual approach of filtering-out their dataset (removing cells with $R^2 < 0.5$) and later for the phase estimation. The authors should utilize multiple and independent methodologies to filter their data by rhythmicity and multiple and independent methodologies for the phase estimation. There are diverse available tools in the circadian literature specifically designed to determine the rhythmicity of circadian signals (e.g. Moore et al., 2014; Thaben and Westermark, 2014; Cenek et al., 2020). This manuscript will strongly benefit from an accurate and state-of-the-art metric of rhythmicity. This is particularly important when using rhythmicity as a method to filter out traces that will not be used to derive the final PTCs.

In view of the reviewer's reservations, we evaluated the rhythmicity and rhythm parameters by 3 different independent methods. The results are summarized in new Supplementary Figure 1a-c, and

clearly indicate that our filter, despite simple, has a very low false-positive error rate. Note that most of the state-of-the-art methods aim to increase power – which is not our top concern. Thanks to the large number of cells, we can be quite stringent with our filter and have only cells with rhythmicity and phase estimates that are accurate beyond doubt.

Detrending

The authors use a moving average (MA) with a 24h time window as detrending strategy. Moving average filters are known to introduce spectral artifacts. The authors could avoid detrending their signal or implement some of the multiple alternative methods to detrend signals that are less prone to spectral artifacts. Using a MA filter is specially concerning when using it with a window size that coincides with the period of the signal of interest (~ 24h). The authors should utilize a filter method less prone to artifacts and/or utilize a window size far from the period of the signal of interest and/or provide the spectral analysis before and after their detrending approach to allow an analysis of how their detrending approach potentially perturbs (or not) the power of the circa 24 hours oscillatory components.

We thank the reviewer for this valuable and valid point. We agree that 24h moving average window is sub-optimal. We explored different windows and compared their reliability, and found that while the 24h window is reasonable, in few cases it created small artifacts that affected the period and phase estimates. We therefore re-run the entire data with a 48h window, which was resilient to this artifact. The revised manuscript includes new figures, which are all based on a 48h window, and overall are very similar to the original figures, hence importantly the new analysis did not affect our conclusions.

PTC reconstruction

1. This manuscript aims to determine the phase shifting effect to a given strength of a perturbation, which then the authors argue that it can be used to understand the dynamics of dose-dependency. In this context it is critical to quantitatively characterize if the phase shift estimated by the authors is only a part of a transient phase shift or a stable phase shift. This could be estimated by obtaining the instantaneous phase post treatment and plot the instantaneous phase-shift as a function of time. If the phase-shift reaches a plateau, then it can be considered a stable phase shift. Otherwise, it is just a partial transient phase estimation limited by the observation time-window.

The reviewer raises valid concerns regarding the stability of the phase shift. Transient phase shift can have two embodiments: (i) the shift is partial and didn't reach its full extent, (ii) the shift is temporary and the oscillator would return to the original phase after few cycles (the latter is the original meaning of the term). To address this concern, we conducted the PTC procedure on several different windows post-intervention (i.e., 1 or 100 nM Dex treatment). As can be observed in new Supplementary Fig. 3e-g, the PTC becomes noisier as we calculated it based on windows further distant from the cue. This is to be expected because of the inherent noise of the single-cell oscillations, as discussed above. As the same happens also in untreated control cells, it is unrelated to the intervention. Importantly, we did not observe a significant change in the extent of the phase shift or the general topology of the PTC when analyzing

different windows. We conclude based on our data and the above analysis that the phase-shift is relatively stable. Eventually, the stochastic drift will desynchronize the population; at the population level this is reflected in amplitude reduction rather than transient phase-shift as known from previous bioluminescence-based experiments (e.g., Nagoshi et al., *Cell* 2004). Notably, most tissue-culture conventional PTC experiments rely on the first or second peak post treatment as the phase estimator (the overall duration of the experiment is limited by the dampening of the whole-population signal). We wish to note that the same stochasticity is the basic feature that CircaSCOPE builds on from first place.

2. Given a common perturbation, an oscillator with small amplitude will suffer stronger phase shifts than an oscillator with a bigger amplitude. In other words, the phase shifting power of a perturbation is strongly affected by the amplitude of the perturbed oscillator. The amplitude of these single-cell circadian oscillators is known to strongly fluctuate in time and to vary from peak to peak. Therefore, to determine the phase-shifting effect of the strength of a perturbation, the authors should estimate the relative amplitude at the time of treatment and not the averaged amplitude before treatment. Moreover, the amplitude of different single-cell oscillators is known to be extremely variable, with some oscillators having ~3X higher amplitude than others. A perturbation normalized to the different amplitudes of the single-cells and normalized to the instantaneous amplitude within a single oscillator is critical for the topological analysis of the PTC and the dynamics of dose-dependency. From the mathematical perspective a phase-shifting perturbation drives the phase across multiple isochrons, and the final phase-shift is determined by the difference between the starting isochron and the final isochron. The number of isochrones crossed (final phase-shift) is a direct consequence of the relationship between the ratio of the oscillator amplitude over the perturbation strength (dose), the direction of the perturbation and the particular isochron spatial structure. In oscillators with strong amplitude fluctuations, the amplitude estimation should be based on an instantaneous amplitude estimation (Guckenheimer, 1975; Gunawan and Doyle, 2006).

The reviewer raises a valid argument regarding the relationship between amplitude, oscillator rigidity and consequently the effect on phase-shifting. Indeed, oscillations with higher amplitude are likely more rigid and confer greater resistance to perturbation and hence weaker phase-shifts. To address this, we measured the amplitude just before an intervention (i.e., 1 or 100 nM Dex treatment) directly from the raw data, so it is instantaneous, and compared it to the observed phase-shift. As depicted in the new Supplementary Fig. 3h, we found no correlation between the two, hence it is unlikely that the amplitude variation is a significant determinant of the PTC topology.

More generally, crossing isochrons is indeed the purest way to describe PTC dynamics. However, determining isochrons experimentally, especially using a single reporter, is extremely challenging. Our method will be nevertheless useful in testing predictions of isochron-based mathematical models regarding specific timing cues, which can be a fertile ground for future studies.

Single cell tracking

The authors record their cells using an Incucyte system seeding their cells in a 24-well plate and imaging with long focal distance 10X objective. With this plate (not the image-lock 96-well plate), the Incucyte system is not capable of perfect stage positioning from one time-point to the next time-point, which means that the cells are not in the same location along the consecutive recordings. This is a known issue, as the authors also acknowledge, that strongly limits single-cell tracking with the Incucyte data. Therefore, the vast majority of Incucyte users do not perform single-cell tracking with these images and only work with population-based analysis. The authors have resolved this challenging problem with a custom-made Cellprofiler pipeline. I consider this a very exciting step and one of the main challenges for readers to be able to implement the method presented in this manuscript. Here, the authors could provide their Cellprofiler pipeline for testing together with a set of movies examples where the single-cell tracking performs as expected so the reader would be able to reproduce the presented results.

We share the reviewer's excitement, as our CellProfiler pipeline is the main image-analysis step that supports the project's efficacy. We provide the CellProfiler pipeline in GitHub, and upon the reviewer's suggestion we uploaded also a sample dataset to Zenodo (doi: 10.5281/zenodo.5139326).

Additional comments

1. Please add page numbers and line numbers to facilitate the review process.

We included page and line numbers in the revised manuscript.

2. The authors claim that their method can estimate phase and period. To determine phase the period calculation is mandatory. In other words, these are not independent capabilities of their approach but a single one.

We removed the sentence in view of the reviewer's reservation.

3. High-throughput: One of the main advantages that the authors claim is that Circa-SCOPE provides a "high-throughput" method, which is in contrast with the classical approach of generating PTCs. The "high-throughput" part lies partially on the fact that these cells are out-of-sync at the time of the perturbation. This is a known observation and expected result of having an uncoupled heterogeneous population of single-cell oscillators and not a specific experimental design from the authors.

The reviewer is absolutely correct, the observation that circadian clocks in cultured cells are uncoupled and exhibit phase heterogeneity is widely accepted and have been observed by many labs for years. We apologize if the manuscript reads as if we are the first to observe it and it was clearly not our intention. We do take advantage of this phenomenon and utilize it to develop an efficient method for PTC extraction. We revised the text accordingly.

4. The authors argue that their method provides a high-resolution PTC. This statement remains unclear to me. If this is related to the 30-min resolution, then the authors should clarify how they obtain a 30min

resolution out of Incucyte recordings performed each 60min. Is this high-resolution is a consequence of fitting and/or interpolation?

We apologize for the inadvertent mistake and thank the reviewer for this note. Indeed, the recording were at 60min resolution and we corrected the statement accordingly.

Reviewer comments, second round- -

Reviewer #1 (Remarks to the Author):

The reviewers have thoughtfully revised their manuscript and it now stands as a significant contribution to the field. Specifically, those interested in cell autonomous clocks will find this manuscript interesting.

Reviewer #2 (Remarks to the Author):

After reading the rebuttal and seeing the new analysis, new figures, the uploaded pipeline, and running an analysis on their raw data (now uploaded), I am very pleased with the authors response and all my concerns are clarified.

There are two comments that I would like to share with the authors. These are minor points but no need to further change the manuscript.

1. The authors insist in their rebuttal that their assumption of stationarity is necessary. To me stationarity is the wrong assumption and also not necessary. What the authors might want is only a robust and simple estimation of the period, so rather than assuming stationarity they could think about their approach in terms of time-averaged period estimation. This would require no changes in their approach and at the same time free them from the assumption of stationarity.

2. They authors mention that they tested a higher-resolution signal analysis method (wavelet approach) but this provided them with too noisy phase estimations and generates a noiser PTC. The simpler case would have been if the authors could show that wavelets generate no improvements in their PTC estimations, and so suggest that the signal is stable enough for their stationary lower-resolution method to work just-well. This was not what they observed but the contrary. Using a higher resolution (time-resolved) method should not generate poorer results. After revising the raw data I rather think their conclusion about "noiser PTC" might be a miss-implementation of their particular wavelet algorithm e.g. how authors defined mother functions, cut-off periods and/or if/how they performed a ridge analysis.

RESPONSE TO REVIEWERS' COMMENTS

We thank the reviewers for their constructive comments and effective process, and happy that they liked the revised piece.

Reviewer #1 (Remarks to the Author):

The reviewers have thoughtfully revised their manuscript and it now stands as a significant contribution to the field. Specifically, those interested in cell autonomous clocks will find this manuscript interesting.

Reviewer #2 (Remarks to the Author):

After reading the rebuttal and seeing the new analysis, new figures, the uploaded pipeline, and running an analysis on their raw data (now uploaded), I am very pleased with the authors response and all my concerns are clarified.

There are two comments that I would like to share with the authors. These are minor points but no need to further change the manuscript.

1. The authors insist in their rebuttal that their assumption of stationarity is necessary. To me stationarity is the wrong assumption and also not necessary. What the authors might want is only a robust and simple estimation of the period, so rather than assuming stationarity they could think about their approach in terms of time-averaged period estimation. This would require no changes in their approach and at the same time free them from the assumption of stationarity.

We see the point; this is another way to look at it. As the reviewer noted, in terms of the manuscript it doesn't imply any further change.

2. They authors mention that they tested a higher-resolution signal analysis method (wavelet approach) but this provided them with too noisy phase estimations and generates a noiser PTC. The simpler case would have been if the authors could show that wavelets generate no improvements in their PTC estimations, and so suggest that the signal is stable enough for their stationary lower-resolution method to work just-well. This was not what they observed but the contrary. Using a higher resolution (time-resolved) method should not generate poorer results. After revising the raw data I rather think their conclusion about "noiser PTC" might be a miss-implementation of their particular wavelet algorithm e.g. how authors defined mother functions, cut-off periods and/or if/how they performed a ridge analysis.

We agree that this is a very interesting aspect for future investigation, as we also noted in the discussion.